



# The variation of particle number size distribution during the rainfall: wet scavenging and air masses changing

Guangdong Niu[1], Ximeng Qi[1,2], Liangduo Chen[1], Lian Xue[1,2], Shiyi Lai[1], Xin Huang[1,2], Jiaping Wang[1,2], Xuguang Chi[1,2], Wei Nie[1,2], Veli-Matti Kerminen[3], Tuukka Petäjä[3], Markku Kulmala[3] and Aijun Ding[1,2]

[1]Joint International Research Laboratory of Atmospheric and Earth System Sciences, School of Atmospheric Sciences, Nanjing University, Nanjing, China.

[2]Jiangsu Provincial Collaborative Innovation Center for Climate Change, Nanjing University, Nanjing, China.

[3]Institute for Atmospheric and Earth Systems Research/Physics, Faculty of Science, University of Helsinki, Helsinki, Finland.

*Correspondence to*: Ximeng Qi (qiximeng@nju.edu.cn)

**Abstract.** Below-cloud wet scavenging is an important pathway to remove atmospheric aerosols. The below-cloud wet scavenging coefficient (BWSC) is the value to describe the ability of rainfall to remove aerosols. The reported BWSCs obtained from the field measurements are much higher than the theory, but the reason for this remains unclear. In this study, based on the long-term field measurements in the Yangtze River Delta of eastern China, we find 28% of the rainfall events are high BWSC events. The high BWSC events show the sudden decrease of particle number concentration in all size bins near the end of rainfall. By investigating the circulation patterns, backward trajectories and the variations of simultaneously observed atmospheric components, we find the cause of the high BWSC events is the air masses changing but not the wet scavenging. The change of air masses is always followed by the rainfall processes and cannot be screened out by the traditional meteorological criteria, which would cause the overestimation of BWSC. After excluding the high BWSC events, the observed BWSC is close to the theory and is correlated with the rainfall intensity and particle number concentrations prior to rainfall. This study highlights the discrepancy between the observed BWSC and the theoretical value may not be as large as it is currently believed. To obtain reasonable BWSCs and parameterization from field measurements, the effect of air masses changing during rainfall needs to be carefully considered.



## 1 Introduction

Atmospheric aerosols have significant impacts on human life by affecting air quality and climate change (Atkinson et al., 2014;Heal et al., 2012;IPCC, 2021;Rosenfeld et al., 2019). The particle number concentration, as well as their size distribution (i.e. particle number size distribution), are the main physical properties that determine the environmental

and climate effects of aerosol particles (Asmi et al., 2011;Chen et al., 2021;Qi et al., 2015;Xausa et al., 2018). The epidemiological studies show that particle number concentration is highly related to human health effects (Chen et al., 2016;Downward et al., 2018;Knibbs et al., 2011). Additionally, only the particles with a size larger than the critical diameter (~50–100 nm) can act as the cloud condensation nuclei, which further alter the cloud properties and thereby the Earth's radiative balance (Kerminen et al., 2012;Schmale et al., 2018;Shen et al., 2019;Twomey, 1991).

The competition between sources and sinks of particles determines the aerosol number concentration, which can be reflected in the variation of particle number size distribution. Many studies focus on the sources of particles while the studies on the sinks of particles are limited (Calvo et al., 2013;Daellenbach et al., 2020;Li et al., 2016;Zhang et al., 2018). Wet deposition is one of the most important sinks for particles (Hou et al., 2018;Textor et al., 2006;Wang et al., 2021), which can be separated into two types, i.e. below-cloud wet scavenging and in-cloud wet scavenging

(Tinsley et al., 2000;Zhao et al., 2015). Earlier studies concluded that below-cloud wet scavenging was negligible compared to in-cloud wet scavenging and thus was not considered in many global models (Stier et al., 2005;Textor et al., 2006). However, for the polluted region, the below-cloud wet scavenging can be the main sink of particles within the planetary boundary layer and should not be ignored (Ge et al., 2021;Xu et al., 2017). The below-cloud wet scavenging is that particles are captured by raindrops through Brownian diffusion, inertial impaction, interception,

thermophoresis and electrical charge effects (Seinfeld and Pandis, 2016). Those below-cloud wet scavenging mechanisms are size-dependent: Brownian diffusion and electrical charge effects are efficient for the particles with size below ~200 nm, while for coarse mode aerosols inertial impaction and interception are the main removal mechanism (Chate and Pranesha, 2004;Greenfield, 1957;Seinfeld and Pandis, 2016). The size-dependent scavenging mechanisms lead to weak below-cloud wet scavenging in the size range of 200-2000 nm, which is the so-called

"Greenfild gap" (Greenfield, 1957). This Greenfild gap is related with the rainfall type, rainfall intensity as well as the aerosol properties (Chate, 2005).

The below-cloud wet scavenging coefficient (BWSC) is the parameter that describes the ability of rainfall to remove particles. There are a number of studies on the BWSCs based on the field observations in different environment (Blanco-Alegre et al., 2018;Chate and Pranesha, 2004;Cugerone et al., 2018;Laakso et al., 2003;Maria and Russell,

2005;Wang et al., 2014;Xu et al., 2019;Zhao et al., 2015;Zikova and Zdimal, 2016). Xu et al. (2019) conducted field measurements in Beijing of China and found the BWSCs calculated by multiple methods are consistent with each other. Zhao et al. (2015) compared the removal effects of thunderstorm rain with those of non-thunderstorm rain and found that non-thunderstorm rain was more effective in removing particles below 500 nm while thunderstorm rain had more effective removal of particles between 500-1000 nm. However, most of the studies have limited datasets

with only few rainfall episodes so that the role of meteorological condition change and instantaneous emission change cannot be ruled out. Some studies collected long-term field measurement datasets to calculate the scavenging





coefficient by using various selection criteria to screen the effects of other factors (Laakso et al., 2003;Roy et al., 2019;Zikova and Zdimal, 2016). Roy et al. (2019) used a long-term field observation dataset to investigate the below-cloud wet scavenging in the eastern Himalaya and found threshold values for the removal effect in terms of rain rate and duration. Laakso et al. (2003) calculated the below-cloud wet scavenging coefficient in boreal forests based on long-term observations of particle number size distribution (PNSD) in Hyytiälä of Finland and analyzed the dependence of BWSC on rainfall intensity. All those studies found the BWSCs calculated from the field measurements were much higher than those from the theoretical estimations and model simulations in the Greenfild gap range. Bae et al. (2010) added electric charging and phoretic effects in the model but still cannot simulate the observed high BWSCs. Until now, the reasons for the large discrepancy between the BWSCs derived from field measurements and theory remain unclear, which limits the simulations of aerosol budget.

Yangtze River Delta (YRD) of eastern China is one of the largest city clusters in the world (Kulmala et al., 2021). Because of rapid urbanization and industrialization, the YRD area suffers from severe air pollution (Ding et al., 2013;Ming et al., 2017;Xie et al., 2015). Meanwhile, located in regions strongly affected by the East Asian monsoon, the rainfall events of various intensities are frequent. Long-term field measurements in the YRD area would provide an ideal opportunity to explore how rainfall removes atmospheric aerosols in polluted environments. In this study, based on the seven-year measurements at the SORPES station in YRD of eastern China, we investigated variations of PNSD during rainfall processes. The aims of this study are: 1) to understand the characteristics of BWSC in polluted environments, 2) to explain the large discrepancy between observed and theoretical BWSC, and 3) to analyze the key factors affecting BWSC.

## 2 Methods

### 2.1 Measurement site and data

The long-term field measurements were conducted at the SORPES station (118°57´10´´E, 32°07´14´´N). The SORPES station is located about 20 km northeast of downtown Nanjing which can be considered a suburban station and is regarded as a regional background station in the YRD region of eastern China (Ding et al., 2013). YRD region is located in the lower reaches of the Yangtze River and has abundant rainfall, especially during the "plum rain season" (June–July). Seasonal variations of monthly accumulated precipitation and rainfall frequency (ratio of the number of monthly rainfall events to the total number of rainfall events) at SORPES from 2012–2018 are shown in Fig. 1a. It could be seen that the accumulated precipitation and rainfall frequency are high in the YRD region, with June-August being particularly prominent. A large number of rainfall events provides sufficient episodes to investigate the below-cloud wet scavenging. In addition, SORPES station is equipped with a number of instruments for measuring the meteorology and atmospheric components, including aerosols, trace gases and relevant meteorological parameters. More details of the SORPES station can be found in Ding et al. (2016).

The PNSDs from 6 nm to 800 nm were observed by the Differential mobility particle sizer (DMPS) (Qi et al., 2015). The time resolution of PNSDs data was 10 minutes and the data from 2012 to 2018 were used. The PNSDs data were





subjected to data quality control and the available data number used in this study was 1793 days. The shaded area in Fig. 1b shows the median PNSD at SORPES station from 2012 to 2018, which demonstrates a multimodal distribution caused by the various sources of particles (Chen et al., 2021;Qi et al., 2015). About 382 rainfall events were recorded

when the PNSD data was available. The red line and green line in Fig. 1b are the median PNSD one hour before and after the rainfall events, respectively. The PNSD before rainfall events was similar to the total median PNSD while the number concentration in each size bin showed a various degree of decrease after the rainfall. The decrease in the particle number concentration after rainfall implies that notable variation of PNSD during the rainfall events and the below-cloud wet scavenging could be a vital sink of aerosols in YRD region.


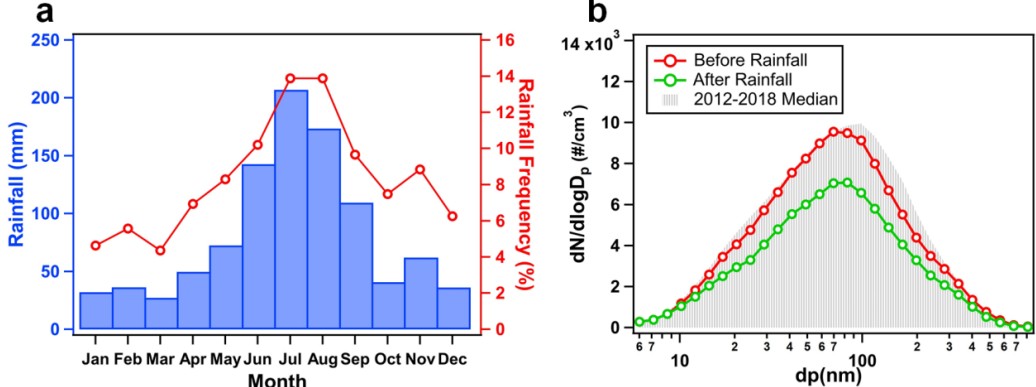

**Figure 1**. (a) Seasonal variations of accumulated precipitation (blue bar) and rainfall frequency (red line) observed at SORPES in 2012-2018. (b) The median PNSD (gray shaded area) in 2012-2018 and 1 hour before (red line) and after (green line) rainfall events.


Besides the PNSD data, this study used the water-soluble inorganic ions data and carbon monoxide data, which were detected by the instrument for Measuring AeRosols and GAses (MARGA) and the trace gas monitor (ThermoTEI 48i), respectively. The meteorological parameters, including rainfall, air temperature, relative humidity (RH), wind speed, and wind direction, were obtained from the weather station (GRWS100) which is installed at the meteorological

observation field of SORPES. In addition to the field measurements at SORPES, reanalysis and satellite retrievals, e.g. the European Centre for Medium-range Weather Forecasts Reanalysis v5 (ERA5) data and the Tropical Rainfall Measuring Mission (TRMM) 3B42 product, were used to support the analysis. The ERA5 data combines weather prediction simulations with observational data to provide accurate hourly meteorological conditions at 0.25°×0.25° spatial grid (Hersbach and Dee, 2016;Olauson, 2018). The TRMM 3B42 product provides precipitation estimates as

a combination of different remote sensors set up in the satellite and has a temporal resolution of three hours and a spatial resolution of 0.25°×0.25°(Huffman et al., 2007).





### 2.2 Rainfall events selection criteria

In order to rule out other impacts except for rainfall on PNSD, all the rainfall episodes need to be screened to select suitable cases. Previous studies used diverse criteria for the selection of rainfall events (Blanco-Alegre et al., 2018;Cugerone et al., 2018;Geng et al., 2019;Laakso et al., 2003;Luan et al., 2019;Pryor et al., 2016;Roy et al., 2019;Wang et al., 2014). Here we summarize all those criteria as follows. Firstly, the rainfall events with sufficient precipitation were selected: (i) the accumulated rainfall no less than 0.4mm, (ii) rainfall intensity no less than 0.3 mm h$^{-1}$, (iii) lasting at least 1 h, and (iv) at least 1-hour interval between each rainfall event. Secondly, as the change of meteorological conditions during rainfall events such as typhoon episodes and frontal passages can affect the particle number concentration, the meteorological selection criteria are needed to remove the effects of meteorological conditions changes: (i) the change in temperature at any adjacent hour during the rainfall events no greater than 6 ℃, (ii) the change in RH at any adjacent hour during the rainfall events no greater than 20%, (iii) the wind speed less than 4 m s$^{-1}$ and (iv) the change in wind direction no more than 90° at the start and end of the rainfall event. According to the above screen criteria, 170 rainfall events among 382 rainfall events were selected. In this study, the selected 170 rainfall events are named as all events in the following discussion if not otherwise specified.

### 2.3 The calculation of below-cloud wet scavenging coefficient

The below-cloud wet scavenging coefficient (BWSC) shows the fraction of particles removed by rainfall in the unit time. The basic equation of variation in the particle number concentration $c(d_p)$ due to rainfall scavenging is described by Eq. (1) (Seinfeld and Pandis, 2016):

$$\frac{dc(d_p)}{dt} = -\lambda c(d_p).$$ (1)

It is assumed that rainfall scavenging varies exponentially with time if there is no chemical reaction or emission during the rainfall event. $d_p$ is aerosol particle diameter and $\lambda(d_p)$ is the size-resolved scavenging coefficient given by Eq. (2)

$$\lambda(dp) = \int_0^\infty \frac{\pi}{4} D_p^2 U_t(D_p) E(D_p, d_p) N(D_p) dD_p,$$ (2)

where $D_p$ is the raindrop diameter, $U_t$ is the falling terminal velocity, $N(D_p)$ is the concentration of raindrops and $E(D_p,d_p)$ is the collision efficiency between the raindrop and aerosol particle. As the collision efficiency, raindrop size distribution as well as raindrop velocity during the rainfall are difficult to be observed, the theoretical scavenging coefficient is actually hard to obtain based on the field measurements.

If rainfall event occurs from $t_0$ to $t_1$, Eq.(1) can be integrated in time as follows (Laakso et al., 2003) :

$$\lambda(dp) = -\frac{1}{t_1 - t_0} \ln\left(\frac{c_1(d_p)}{c_0(d_p)}\right).$$ (3)





In Eq. (3), $c_0(d_p)$ and $c_1(d_p)$ are the median particle number concentrations in each size bin before the rain occurs ($t_0$) and after the rainfall ends ($t_1$), respectively. In this study, we use Eq. (3) to calculate the scavenging coefficient and compared it with the theoretical estimation.

**3 Results and Discussions**

**3.1 High BWSC events observed in eastern China**

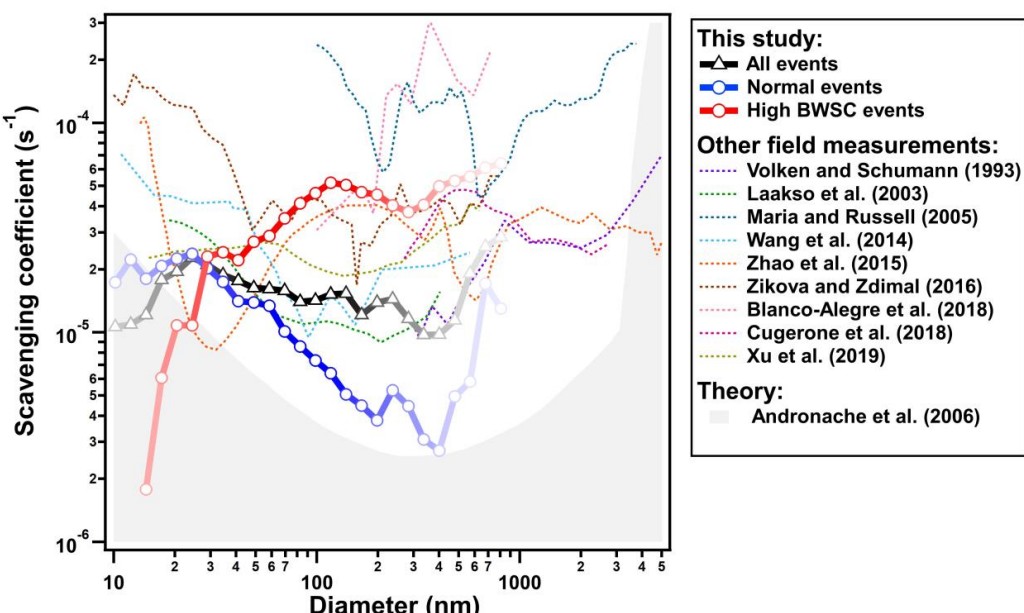

**Figure 2**. Below-cloud wet scavenging coefficients at SORPES station and the comparisons with other studies. The
lines of this study are color coded with the particle number concentration in the corresponding size bin (color from light to dark represents the number concentration from low to high).

The BWSCs calculated in this study as well as the comparisons with other studies are shown in Fig. 2. The black line is the BWSC of all events at SORPES in YRD of eastern China with color shade representing the particle number concentration in the corresponding size bins. The BWSC of all events at SORPES reaches a minimum in the 70-500 nm size range, but it is much higher than the theoretical value, which is similar to the results in Beijing, Mount Huang, Prague and southern Finland (Laakso et al., 2003;Wang et al., 2014;Xu et al., 2019;Zikova and Zdimal, 2016). Other studies observed higher BWSCs relative to theoretical value as well, although the BWSCs differ from this study to some extent (Blanco-Alegre et al., 2018;Cugerone et al., 2018;Maria and Russell, 2005;Zhao et al., 2015;Volken and
Schumann, 1993). In general, most of the BWSCs obtained from the field measurements have 1-2 orders of magnitude





higher than the theory (Andronache et al., 2006;Wang et al., 2010). It is often considered to reduce this gap by adding extra below-cloud wet scavenging mechanisms to the theory. Bae et al. (2010) added electric charging and phoretic effects in the theoretical calculation, which made the discrepancy with measurements less. However, even with the additional mechanisms of below-cloud wet scavenging considered, the discrepancy between observation and theory

is still large (Bae et al., 2010). Wang et al. (2010) proposed that the additional physical processes, such as cloud-aerosol microphysics, turbulent transport and mixing, could be the reasons for the large discrepancy between observed and theoretical BWSCs. In this study, we aim to understand this discrepancy from the perspective of the synoptic processes and air masses changing.

**Table 1.** BWSCs in corresponding particle size range observed in this study and other studies.

| Size range (nm) | BWSC ($s^{-1}$) | Note | Source |
|---|---|---|---|
| 10-800 | $9.74 \times 10^{-6}$-$2.84 \times 10^{-5}$ | All events | This study |
| 10-800 | $2.72 \times 10^{-6}$-$2.37 \times 10^{-5}$ | Normal events | This study |
| 10-800 | $-8.15 \times 10^{-6}$-$6.4 \times 10^{-5}$ | High BWSC events | This study |
| 10-510 | $7 \times 10^{-6}$-$4 \times 10^{-5}$ | Long-term study | Laakso et al., 2003 |
| 10-800 | $3.1 \times 10^{-5}$-$1.5 \times 10^{-4}$ | Long-term study | Zikova & Zdimal, 2016 |
| 13-750 | $1.08 \times 10^{-5}$-$7.58 \times 10^{-4}$ | Case study | Chate & Pranesha, 2004 |
| 100-4000 | $4.41 \times 10^{-5}$-$2.39 \times 10^{-4}$ | Case study | Maria & Russell, 2005 |
| 10-500 | $9.43 \times 10^{-6}$-$7.08 \times 10^{-5}$ | Case study | Wang et al., 2014 |
| 10-10000 | $7.48 \times 10^{-6}$-$7.46 \times 10^{-4}$ | Case study | Zhao et al., 2015 |
| 250-3000 | $2 \times 10^{-5}$-$5 \times 10^{-5}$ | Case study | Cugerone et al., 2018 |
| 14-740 | $1.86 \times 10^{-5}$-$4.1 \times 10^{-5}$ | Case study | Xu et al., 2019 |

Based on the variation of the particle number size distribution during the rainfall, we separated the well-screened rainfall events into two types: the normal scavenging events and high BWSC scavenging events. The particle number concentrations in normal scavenging events (122 events, 72% of total events) showed a moderate decreasing trend

over time (Fig. 3a). During the rainfall, the number concentration of small particles is first to decrease, after which the scavenging phenomenon gradually shifts to the size around 100 nm (Fig. 3a). It is consistent with the theory that the BWSC below 300 nm decreases as the particle size increases (i.e. the so-called "Greenfild gap"). Different from the normal scavenging events, the high BWSC events (48 events, 28% of total events) showed a rapid decrease of particle number concentration in all size ranges near the end of the rainfall (Fig. 3b). The BWSCs of high BWSC events reach





a maximum between 70-300 nm, which largely exceeds the theoretical values as well as most of the previous observational studies. The extremely high BWSC values and a certain number of high BWSC events lead to the overestimation of BWSC in the total events (Fig. 2). Table 1 compares the BWSCs in this study with other studies. In general, the normal scavenging events in this study had much lower BWSC than that in other studies while the BWSC for all events is comparable with other studies. The BWSC becomes much closer to the theoretical value when high

BWSC events are excluded. In the following sections, we will furtherly analyze the reasons for the high BWSC events.

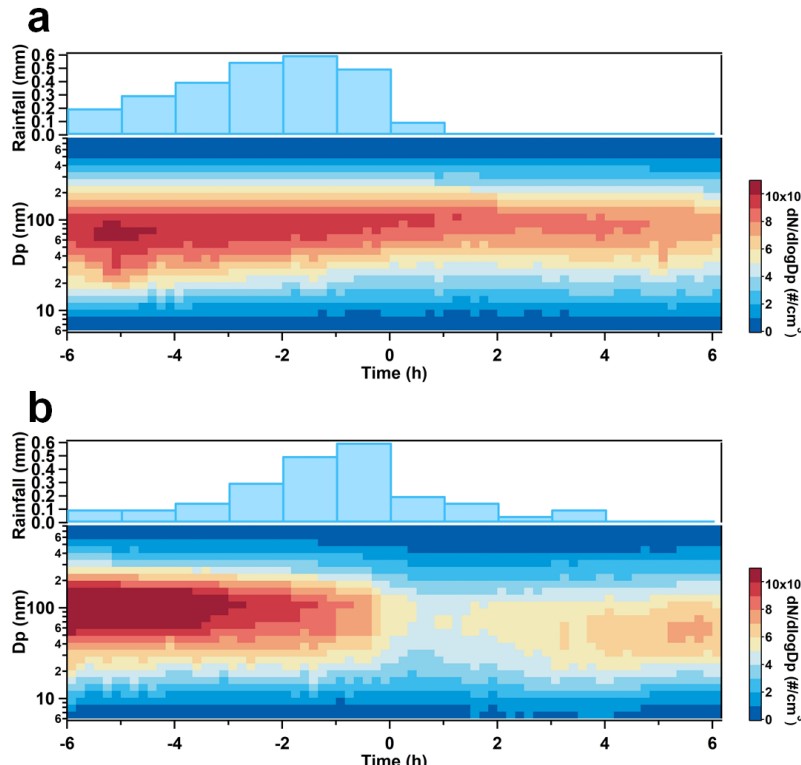

**Figure 3**. The variation of PNSD and precipitation for (a) normal scavenging events (the end of rainfall is marked as 0) and (b) high BWSC events (the time when the particle number concentration rapidly decrease is marked as 0).


### 3.2 Synoptic processes during high BWSC events

To investigate the reasons for the high BWSC events, we analyzed the synoptic processes during these events. Through the subjective classification, the synoptic situation of high BWSC events can be divided into four main categories, i.e. westerly trough type (16.6%), stationary front type (31.3%), vortex weakening type (22.9%) and vortex type (29.2%).





The westerly trough type, stationary front type and vortex weakening type have clear movement of the synoptic systems, while vortex type shows relatively stable circulations during the rainfall.

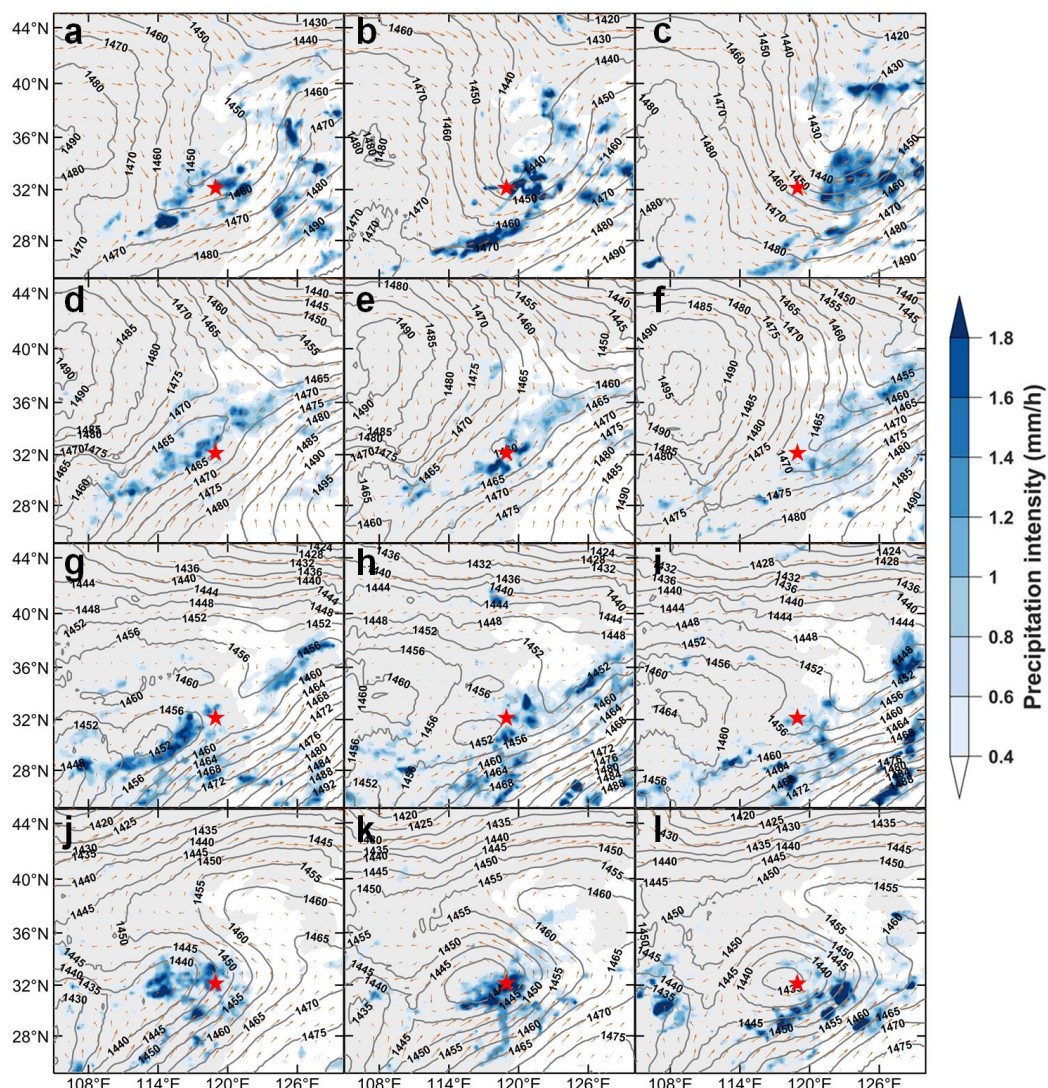

**Figure 4**. Distributions of winds and geopotential heights at 850 hPa and precipitation intensity before, at, and after
the moment of sudden decrease in particle number concentration for (a-c) westerly trough type, (d-f) stationary front type, (g-i) vortex weakening type, and (j-l) vortex type. The red pentagram shows the location of SORPES station.



i) Westerly trough type

The rainfall events of this type are caused by the westerly trough, occurring mostly in spring and summer. Figures 4a-c show that the trough moved from west to east during the rainfall events. The rain belt is commonly located to the east of the westerly trough because this region is dominated by surface convergent winds, low pressure and ascending motion (Wallace and Hobbs, 2006). For rainfall events observed at SORPES station, the SORPES station was located east of the westerly trough before the moment of the sudden decrease in particle number concentration, and shifted to the west of the trough when the rainfall event was about to end. To further understand the transport characteristics of the air mass, we calculated the 3-day backward trajectories of air masses using Hybrid Single-Particle Lagrangian Integrated Trajectory (HYSPLIT) model (Draxler and Hess, 1998) . Figure 5a shows the backward trajectory of air masses arriving at SORPES station before, at and after the moment of sudden decrease in particle number concentration. Consistent with the movement of the westerly trough as shown in Fig 4a-c, the backward trajectories of air masses had a clockwise rotation during the rainfall event. Towards the end of the rainfall event, the air masses were mostly from the rainfall areas and have low particle number concentrations. This air mass transition explains the sudden decrease in particle number concentration for all the size bins. A typical case of type i high BWSC event as well as the circulation evolvements and backward trajectories are shown in Figures S1-S3.

ii) Stationary front type

The cold front events can be screened out by the meteorological criteria as described in section 2.2. However, the stationary front events, which occurred mostly in summer and autumn, cannot be excluded due to the slow movement of the synoptic systems. Figures 4d-f show that the stationary front moved slowly from northwest to southeast during the rainfall events at SORPES station. The warm air mass climbed along the front to the cold side, accompanied by a drop in temperature, and the water vapor to start condensing (Wallace and Hobbs, 2006). Therefore, rain belt is commonly near the front on the side of the cold air and moves with the movement of the front. The SORPES station was located on the warm side of the front before the rapid decrease of particle number concentration while on the cold side of the front after that. Consistent with the movement of the stationary front (Fig 4d-f), the backward air masses of SORPES station switch from the YRD region to the northern area of station. Towards the end of the rainfall event, the air masses are mostly from the rainfall areas and have low particle number concentration. A typical case as well as the circulation evolvements and backward trajectories are shown in Figures S4-S6.

iii) Vortex weakening type

The rainfall events of this type are caused by a small trough, occurring mostly in summer and autumn. Figures 4g-i show that the vortex weakens into the small trough and moved from west to east during the rainfall events at SORPES station. The rainfall areas are usually scattered ahead of the small trough (i.e. to the east of the trough in the westerly zone) because of the strong ascending motion in this area. The SORPES station was located ahead of the small trough during the rainfall event, and shifted to the back of the small trough when the rainfall event was about to end. Unlike the first two types, backward air masses of this type did not present horizontal rotation (Fig. 5c). However, when the rainfall is about to end, the origin of air mass gradually shifted from the ground to the high altitude due to the control of the descending motion air masses. Since at high altitude the particle number concentrations are generally low (Qi




et al., 2019), changes in the origin of air masses explain the sudden drop in particle number concentration. A typical

case as well as the circulation evolvements and backward trajectories are shown in Figures S7-S9.

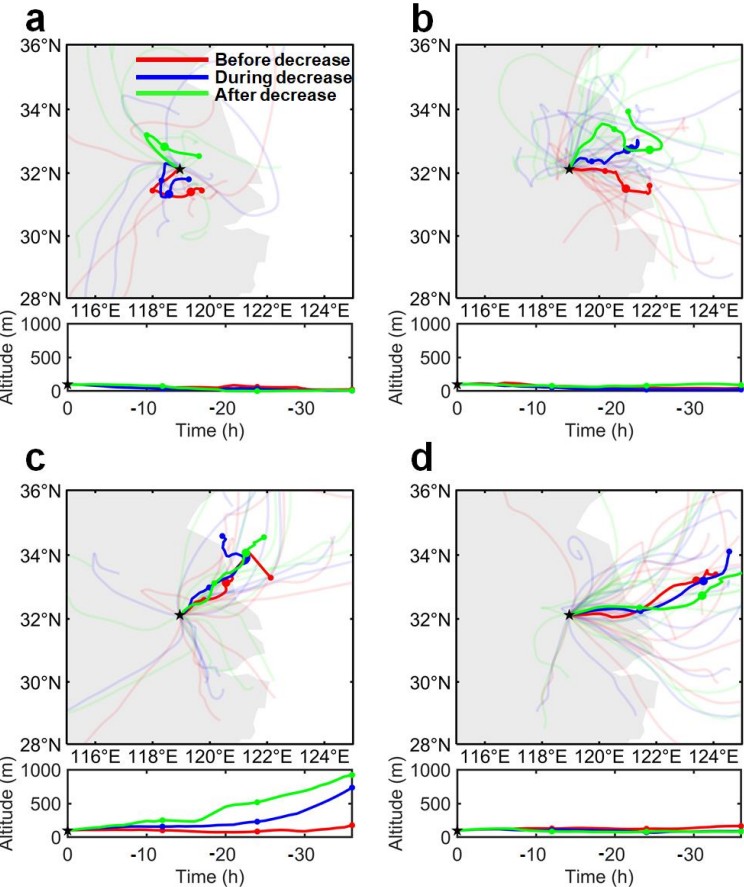

**Figure 5**. The backward trajectory before, at, and after the moment of sudden decrease in particle number concentration for (a) westerly trough type, (b) stationary front type, (c) vortex weakening type, and (d) vortex type.


iv) Vortex type

The rainfall events of this type are caused by the vortex, occurring mostly in spring and summer. The synoptic systems of the vortex type show no clear movement, which is different from other types. As shown in Fig 4j-l, the SORPES station is located on the eastern side of the vortex. The rainfall area moves from the western side of the SORPES

station to the eastern side and covers the whole YRD region during the rainfall event. The air masses for vortex type are mostly from the East China Sea and pass through the rain belt in the YRD region. Although the backward





trajectories of air masses have no clear horizontal rotation or vertical height change, the properties of air masses may still vary during the rainfall event. A typical rainfall event of vortex type as well as the synoptic situation and backward trajectories are shown in Figures S10-S12. The particle number concentration starts to decrease when the rainfall

occurs but have a significant decrease in all the particle size, accompanied by the drop in carbon monoxide concentration during the rainfall. The simultaneous changes in the particle number concentration and carbon monoxide concentration indicate the sudden decrease of particle number concentration is caused by the site being controlled by the marine air masses with minor influence from anthropogenic emissions in YRD.

In conclusion, we find that the change of air masses is the main reason for the sudden decrease in particle number

concentration during high BWSC events. It is worth to note that the transition of air masses is usually accompanied by changes in rainfall processes, e.g. the end of precipitation, which could easily mislead the calculation of BWSCs and cause the calculated BWSC much higher than the theory. Although the filtering procedures (see section 2.2) can screen some cases of significant air masses changes (e.g. cold front events, typhoon events), many more air masses transition cases with no obvious variations in meteorological parameters may not be screened.

**3.3 The comparisons between high BWSC events and normal events**

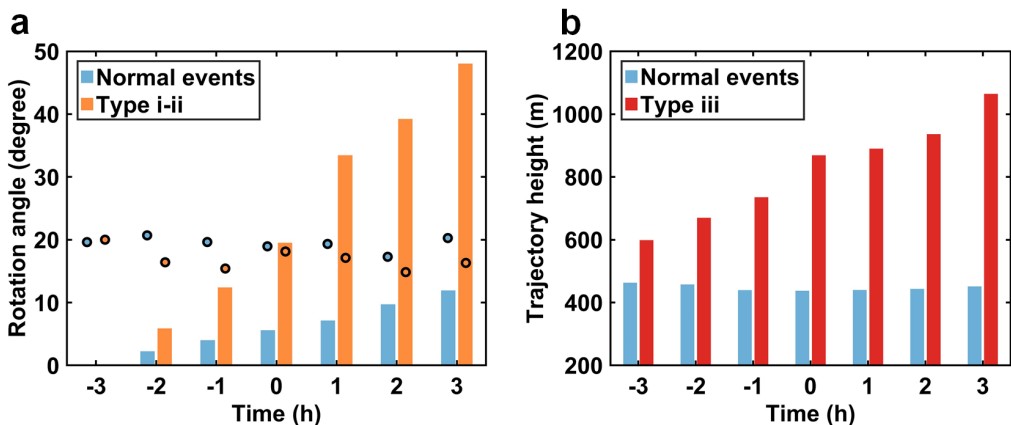

**Figure 6**. (a) The rotation angle of backward trajectory (bars) and wind direction (dots) during rainfall for normal event (blue) and type i-ii high BWSC events (orange). (b) The trajectory height during rainfall for normal event (blue)

and type iii high BWSC events (red).

To compare the high BWSC events with the normal events, we investigated the large-scale circulation patterns of the normal events as well. Through the subjective procedure, the circulation patterns of normal events could be mainly divided into three types, i.e. trough type (14.8%), vortex weakening type (34.4%) and vortex type (50.8%) (Fig. S13).

Compared to high BWSC events, the circulation patterns for normal events does not change much during the rainfall. Consistent with the circulation pattern, the backward trajectories of air masses for the three types of normal events





show minor variations during the rainfall (Fig. S14). Figure 6a shows the comparisons of the rotation angle of air mass trajectory and wind direction between type i-ii high BWSC events and normal events during rainfall. The air mass trajectory rotation angle of type i-ii high BWSC events varied more drastically than normal events, while the wind direction did not vary much in both types. It indicates that significant changes in the backward trajectories of air masses are not reflected in the wind direction observed at ground level, which explains why the type i-ii high BWSC events were not excluded by the selection criteria described in section 2.2. Figure 6b shows the differences in air masses trajectory height between type iii high BWSC events and the normal events. The backward air masses trajectory height increases significantly in type iii high BWSC events, while it barely changes in the normal events. In general, the relatively stable circulation evolvements during the normal events cause little changes in air masses, so that the decrease of particle number concentration could be mainly attributed to the below-cloud wet scavenging.

The atmospheric components observed at SORPES station can furtherly support that the origin of air masses changes during the high BWSC events. Figure 7a shows the differences of carbon monoxide concentration between high BWSC events and normal events. There is no significant change in carbon monoxide concentration for normal events. As carbon monoxide is insoluble in water, the wet scavenging can hardly induce significant changes in carbon monoxide. However, carbon monoxide concentration decreases significantly during the high BWSC events, suggesting the changes of air masses. Figures 7b and 7c show the variations of the proportion of sulfate and nitrate in water-soluble ions (WSI) during high BWSC events and normal events. Similarly, the proportion of sulfate and nitrate in WSI changed indistinctively for normal events while it changed significantly during the high BWSC events. The proportion of sulfate increased and the proportion of nitrate decreased during the rainfall for the high BWSC events. To be concluded, in order to investigate the below-cloud wet scavenging based on field measurements, a detailed analysis of circulation patterns, air mass origins and atmospheric multi-components is required to select the cases that are mainly affected by the below-cloud wet scavenging.

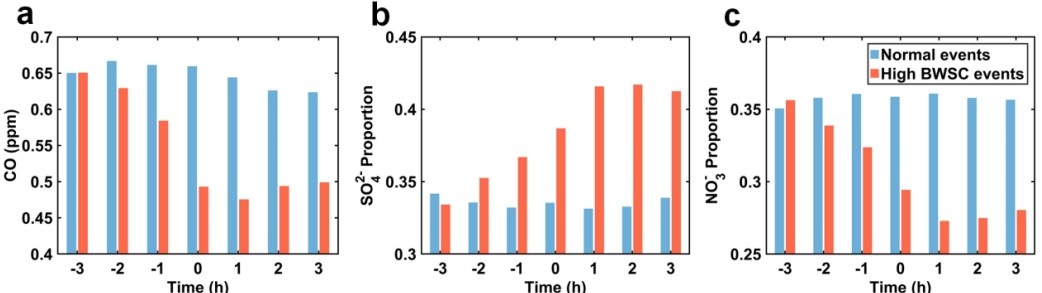

**Figure 7**. The comparisons of the variations of (a) CO concentration, (b) the proportion of sulfate in WSI and (c) the proportion of nitrate in WSI during normal scavenging events (blue) and high BWSC events (red).

**3.4 Factors influencing BWSCs**




Previous studies found a close relationship between the rainfall intensity and BWSC and gave the parameterization
       between them (Blanco-Alegre et al., 2018;Chate and Pranesha, 2004;Laakso et al., 2003;Volken and Schumann,
       1993;Wang et al., 2014;Xu et al., 2019). Figure 8a shows the relationship between the BWSC and rainfall intensity
       for high BWSC events and normal events observed at SORPES. As the scavenging of the high BWSCs events is
       mainly caused by the change of air mass but not the wet deposition of raindrops, the BWSC for high BWSC events

was not affected by the rainfall intensity. For normal scavenging events, the BWSC increased rapidly with rainfall
       intensity until the rainfall intensity reaches 1 mm/h, while at the rainfall intensity greater than 1 mm/h, the rate of the
       increase of BWSC with rainfall intensity slows down. This is because the BWSC is limited by the rainfall intensity
       under the low rainfall intensity condition while it switches to the limitation of background particle number
       concentration under high rainfall intensity conditions. The rainfall intensity turning point between two limitation

intervals increases with the elevation of particle number concentration, which is around 1.5 mm/h at high number
       concentration and around 0.5 mm/h at low number concentration (Fig. 8a).

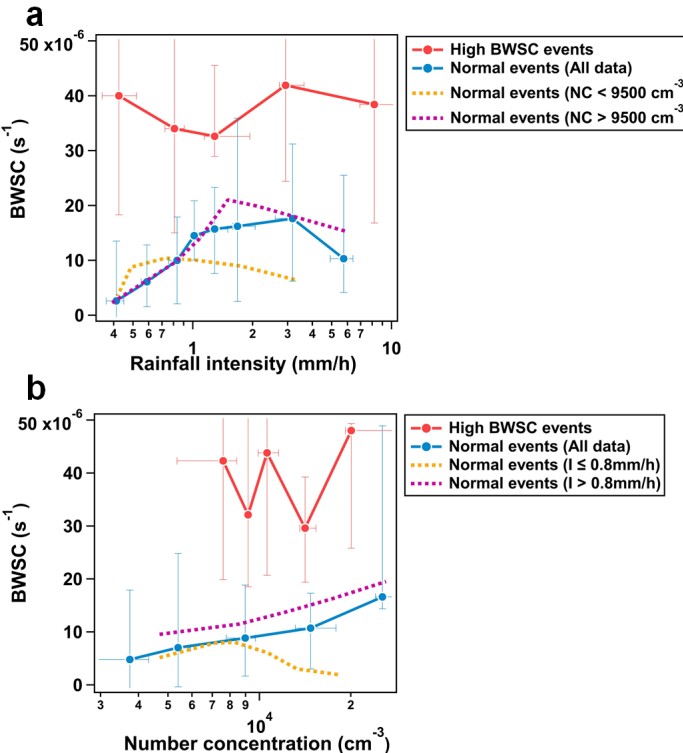

**Figure 8**. Below-cloud wet scavenging coefficients as the function of (a) rainfall intensity and (b) particle number
concentration prior to rainfall events. The dots represent the median and the error bars represent the upper and lower
       quartiles.



The particle number concentration prior to the rainfall events represents how many particles in ambient can be scavenged by rain droplets. As shown in Fig. 8b, for normal scavenging events, there is a significant positive

correlation between the BWSC and particle number concentration prior to the rainfall events, especially when the rainfall intensity is high. It indicates that the collisions between the rain droplets and aerosol particles are more efficient at high particle number concentrations. Moreover, when the particle number concentration is high, the coagulation between aerosol particles could also lead to a decrease in the particle number concentration, which will affect the estimation of the BWSC. Therefore, when parameterizing the BWSC with rainfall intensity based on field

measurements, we need to not only carefully select the scavenging cases that are really caused by the wet scavenging (i.e. normal scavenging events in this study), but also take the particle number concentrations prior to the rainfall events into account.

## 4 Conclusions

Based on the long-term field measurements at SORPES station in the Yangtze River Delta region of eastern China, we investigated the below-cloud wet scavenging of submicron particles in different size ranges in polluted environments. We find that 28% of total rainfall events are high BWSC rainfall events, which commonly show sudden decreases of particle number concentration in all size ranges near the end of rainfall events. The high BWSC rainfall events cannot be screened out by the traditional filtering procedures based on meteorological parameters, which could

contribute to much higher observed BWSCs due to the certain numbers of events. By investigating the circulation evolvements, air mass origins and the variation of the observed atmospheric components, we find the cause of high BWSC rainfall events is the air masses changing. The changes of air masses origins are accompanied by the end of rainfall events, which could easily mislead the calculation of BWSCs. The BWSCs for high BWSC events are not correlated with the rainfall intensity, which could then affect the parameterization of BWSC. After excluding the high

BWSC events, we find the BWSC is well correlated with both the rainfall intensity and the particle number concentrations prior to the rainfall events.

This study highlights the discrepancy between the observed BWSC from field measurement and the theoretical value may not be as large as it is currently believed. When calculating the BWSCs based on field measurements, the impacts of air masses changing need to be carefully excluded. Long-term observations on multiple atmospheric components

can help us to obtain the BWSCs that are truly caused by the below-cloud wet scavenging and update the parameterization in the numerical models. Moreover, rainfall events with specific circulation patterns could cause air masses changes. Thus, this study highlights accurate descriptions of the evolution of synoptic systems as well as the wet scavenging processes are equally important for simulating the air pollution in numerical models.



*Author contributions.* XQ and AD conceptualized the study. GN made the main data analysis with the supervision from XQ and AD. GN, LC, JW and XC conducted the measurements. GN and XQ wrote the original manuscript with contributions from all co-authors.

*Competing interests.* The contact author has declared that none of the authors has any competing interests.


*Acknowledgments.* This work was supported by the National Natural Science Foundation of China (42175113), the Fundamental Research Funds for the Central Universities (14380191) and the Research Funds for the Frontiers Science Center for Critical Earth Material Cycling, Nanjing University.

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
