# Peer review of "The variation of particle number size distribution during the rainfall: wet scavenging and air masses changing"

_EGUsphere, 2023_

## Referee Comment (RC2)

**Review comments to manuscript acp-2023-33**

**The variation of particle number size distribution during the rainfall: wet scavenging and air masses changing**

**by Guangdong Niu et al.**

**Review Comments**

The paper investigates possible sources of discrepancy in the below-cloud wet scavenging coefficient (BWSC), a metric related to the ability of rainfall to remove suspended particles, between observations and theoretical results. Past studies have highlighted that the BWSC inferred from measured particle number size distributions (PNSD) are larger than the theoretical ones, but sources of these discrepancies remain unclear. This paper uses long-term (i.e., 7 years, ~1800 days after data quality control) PNSD data collected at the SORPES station in eastern China to quantify BWSC in a polluted atmosphere and to investigate sources of discrepancies between observed and theoretical BWSC. Auxiliary to the analysis is the ERA5 and TRMM data that provide meteorological and rainfall information over the area, respectively. The authors select rainfall days above a certain intensity and duration and exclude extreme events and frontal passages that will significantly impact PNSD. This leads to 170 events being analyzed. Two categories of events are identified based on BWSC (high and normal events). Events are also classified based on changes in synoptic systems that may lead to a different role played by the BWSC. Pollutant concentrations (e.g., CO) are also used as tracers to track changes in air masses. The manuscript is well written, the results are innovative, and topic is of high relevance for Atmospheric Chemistry and Physics. However, the following concerns should be addressed to be considered suitable for publication.

**General comments:**

- Few results are shown for the normal scavenging vs high scavenging events in terms of PNSD. Two specific events are shown in Fig 3, but it would be interesting to know more about the overall statistics. How do the two distributions differ on average? What is the variability/uncertainty in the PNSD and rainfall distribution over time for these two types of events over the analyzed period?
- More details should be provided in the methods on how the different events are classified in terms of BWSC and also in terms of the synoptic driving systems. Also for example, in Figure 6 it's not clear what the trajectory height means and how it is computed. Rainfall events may be synoptic driven or more locally generated. How are these differences accounted for?
- The dependence of BWSC on the pre-existing particle number concentrations is very interesting, but more details are needed. Does this relationship vary as a function of the particle size as well or is it constant from the smallest to the largest diameters? Does the pre-existing PNSD play a role in dictating BWSC or is the total particle number the key driver?
- How generalizable these results are? The authors could contextualize the emission regimes, the typical PNSD, the type of air mass changes, with those observed in other studies and comment on which regions the BWSC could be impacted by some of the identified drivers.

---

## Author Comment (AC1)

**Response to Reviewer #1**

Overall, this manuscript makes a nice contribution to understanding observations of below-cloud wet scavenging and the need to remove air-mass changes. It deserves to be published. However, I have some comments that I'd like to be addressed. I'm particularly concerned about Figure 8 and its interpretation, and this is the focus of my last 4 comments.

**Response:** Thanks for providing helpful and constructive comments, which help us further improve the manuscript. The changes in the revised manuscript were highlighted in red color. Below is our point-by-point response to each comment.

Comment 1:

The manuscript needs a general review of English (including title and throughout). Generally, the issues don't prevent the points from being understood, but things can be cleaned up to make the manuscript a bit easier to read. As an example with the title, it would be more common to say, "The variation of the particle number size distribution during the rainfall: wet scavenging and air masses changing".

**Response:** Thanks for the comment and suggestion. We have conducted a general review of English through our co-authors, which has made the manuscript more readable. Please see the revised version for details.

Comment 2:

Section 2.2: The criteria listed here have somewhat arbitrarily chosen cutpoints. How sensitive are the results to changes in these cutpoints?

**Response:** Thanks for the comment. We reviewed the literatures related to below-cloud wet scavenging studies based on field measurements. In the articles that specifically describe the selection criteria for rainfall events (Blanco-Alegre et al., 2018;Cugerone et al., 2018;Geng et al., 2019;Laakso et al., 2003;Luan et al., 2019;Pryor et al., 2016;Roy et al., 2019;Wang et al., 2014), there is no uniform standard in the selection criteria. The relevant cutpoints in this study were chosen based on the existing cutpoints in the literature mentioned above.

To test the sensitivity of the cutpoints to the results, we have reduced all cutpoints of the meteorological selection criteria by 20%, which makes the screening condition stricter. The strict meteorological selection criteria is: (i) the change in temperature at any adjacent hour during the rainfall events no greater than 4.8 °C, (ii) the change in RH at any adjacent hour during the rainfall events no greater than 16%, (iii) the wind speed less than 3.2 m s$^{-1}$ and (iv) the change in wind direction no more than 72° at the start and end of the rainfall event. The number of events inevitably decrease (normal events from 122 to 91, high BWSC events from 48 to 30). Nevertheless, the changes in BWSC after using strict cutpoints is not significant compared to the previous one (Fig. R1). Therefore, the chosen cutpoints don't affect the major results of this study. Considering the referee's comments, we have added the discussions on the selection of cutpoints in section 2.2. Please see Line 131-132 and Line 142-143.

[Figure]

**Figure R1.** Below-cloud wet scavenging coefficients at the SORPES station for the (a) strict meteorological selection and (b) original selection criteria.

Comment 3:
Figure 2: I found it hard to see the "theory" curve on the plot (the pink was very light), please make it darker or make it a line.
**Response:** Thanks for the comment and suggestion. We have made the "theory" curve of the Fig. 2 darker in the revised version. Please see Figure 2.

Comment 4:
Figure 2: What assumptions were made for the "theory" curve? The rainfall rate and the rain size distribution will affect this curve. Related, the rain conditions for each of the obs studies on the plot may different, so it's worth discussing these potential differences.
**Response:** Thanks for the comment and suggestion. The assumptions of the "theory" curve are from Andronache et al. (2006). Raindrop-particle collection efficiency is from Slinn (1983) with the phoresis and electric forces taken into account; raindrop number size distribution is Marshall-Palmer distribution; raindrop terminal velocity is from Atlas and Ulbrich (1977); rainfall intensity is 10 mm/h. The rainfall rate and the rain size distribution do affect the "theory" curve. However, the differences between the curves obtained from different theoretical calculation is relatively small compared to their differences with field measurements (Wang et al., 2010). The rain conditions for each previous observation studies are different. In general, the stronger rainfall intensity lead to higher observed scavenging coefficient.

Considering the referee's comments, we have added more descriptions of the assumptions of the "theory" curve in the revised manuscript. Please see Line 171-173. And the relevant discussions on the rain conditions in each observation study are added to the revised manuscript. Please see Line 181-182 and Table1.

Comment 5:
L241-242: I don't understand this statement. It might be because I'm not sure what a

"backwards air mass" is (do you mean "back trajectory"?).

**Response:** Thanks for the comment. The word "backward air mass" is not clear enough here and can be misunderstood. We have replaced "backward air mass" with "backward trajectory" in the revised version. Please see Line 253.

Comment 6:

Figure 6: Is there a reason why type iv is not plotted here? Should mention why.

**Response:** Thanks for the comment and suggestion. The type iv's backward trajectories have no clear horizontal rotation or vertical height change (Fig. 5d). Therefore, we didn't add the type iv to Fig. 6. Considering the referee's comments, we have added the description in the revised manuscript. Please see Line 296-298.

Comment 7:

Figure 7: The CO change here seems like the most straightforward evidence of the airmass change, given that it's a non-scavenged species. Not necessary, but this could be made more prominent in the paper (e.g., abstract, moving it forward in the results, etc.).

**Response:** Thanks for the suggestion. We have emphasized the characteristic of the CO change in the abstract and conclusion. Please see Line 19-20 and 386-388.

Comment 8:

Figure 8: What size are these BWSC coefficients for? Is it for the total number of particles, regardless of their size? If true, this complicates the interpretation of the figure since there is likely a correlation between the particle number concentration and the size of the particles. I recommend instead making this plot for the BWSC for a specific size, which makes for a fairer comparison (or at least having this as a second panel).

**Response:** Thanks for the comment and suggestion. The BWSCs in Fig. 8 are the median over particle size range from 10 nm to 500 nm. In the revised manuscript, we presented the BWSCs for each size bin (Fig. R2c, d).

As shown in Fig. R2c, the dependence of BWSC on rainfall intensity for particles below 100 nm (i.e. ultrafine particles) is similar with that of total particles, since the ultrafine particles dominate the particle number concentration at SORPES. The BWSCs are low for the particles larger than 100 nm due to the exist of "Greenfield" gap (Greenfield, 1957). Nevertheless, the increase in BWSC with increasing rainfall intensity can also be found for the particles larger than 100 nm. The dependence of BWSC on particle number concentration prior to rainfall event is obvious for ultrafine particles as well (Fig. R2d). In general, although the BWSCs are related to the particle size, the dependences of BWSCs on rainfall intensity as well as particle number concentration is uniform over all the size bins.

Considering the referee's comments, we have added the BWSCs in each size bin in Fig. 8 and the discussions of BWSCs for different size bins in the revised version. Please see Figure 8 and Line 357-365.

[Figure]

**Figure R2.** Below-cloud wet scavenging coefficients (median over particle sizes 10-500 nm) for high BWSC events and normal events as the function of (a) rainfall intensity and (b) particle number concentration prior to rainfall events. The dots represent the median and the error bars represent the upper and lower quartiles. Below-cloud wet scavenging coefficients in each size bin for normal events as the function of (c) rainfall intensity and (d) particle number concentration prior to rainfall events.

Comment 9:

L327-329: It does not make physical sense that the BWSC depends on number concentration. Below-cloud wet scavenging is a 1st-order loss process, particles should not be influencing other particles' ability to be scavenged. It seems much more likely that particle number and the average size of the particles are at least somewhat correlated, and that is driving the relationship here. This is why making Figure 8 show the BWSC for a specific size would be easier to interpret. (Also, is there a relationship between rain rate or the rain drop size distribution and the number of particles? A correlation here seems less likely than the relationship between particle number and particle size, but worth checking since it could also influence the interpretation of Figure 8.)

**Response:** Thanks for the comment and suggestion. As shown in Fig. R2d, the BWSCs increase with the increasing particle number concentration prior to rainfall event in most of size bins. Theoretically, below-cloud wet scavenging is a 1st-order loss process and should not have the relationship with the particle number concentration prior to rainfall event. The basic equation of variation in the particle number concentration $c(d_p)$

due to rainfall scavenging is described by Eq. (1) (Seinfeld and Pandis, 2016):

$$\frac{dc}{dt} = -\lambda c \qquad (1)$$

$\lambda$ is the scavenging coefficient (i.e. BWSC). As shown in Fig. R3a, the particle number concentration varies exponentially with time if no other processes (other sources or sinks of particles) are present during the rainfall event. The slope of the line is the scavenging coefficient ($\lambda$) and does not vary with the particle number concentration prior to rainfall ($c_0$).

However, in the real ambient environments, other processes such as particle formation, primary emissions, coagulation, etc. cannot be excluded. For example, if there is a stable source of particles, the variation of particle number concentration can be described by:

$$\frac{dc}{dt} = -\lambda c + A. \qquad (2)$$

A is the formation rate of the particles due to the stable source. The variation of particle number concentration based on Eq. (2) is shown in Fig. R3b by assuming consistent BWSC and formation rate.

In field measurements, it is common to use Eq. (3) to calculate the BWSC:

$$\lambda = -\frac{1}{t_1 - t_0} \ln\left(\frac{c_1}{c_0}\right). \qquad (3)$$

$c_0$ and $c_1$ are the median particle number concentrations before ($t_0$) and after ($t_1$) the rainfall event, respectively. Assuming same duration of rainfall event, the BWSCs can be calculated based on Eq. (3) for different initial particle number concentration (dash lines in Fig. R5b). Although the actual BWSC is consistent, the calculated BWSC can increase with the increasing initial particle number concentration. Therefore, the relationship between BWSC and particle number concentration prior to the rainfall event can be caused by other processes except for below cloud wet scavenging. In real ambient environments, other processes perturb the calculation of BWSC when using the BWSC calculation method as described by Eq. (3). We thereby highlight the need for further research on BWSC calculation methods based on field measurements.

[Figure]

**Figure R3.** The variation of particle number concentration with time during rainfall under theoretical conditions (a) without and (b) with the stable source. (Assuming the scavenging coefficient ($\lambda$) of $1\times10^{-5}$ s$^{-1}$ and the particle formation rate (A) of $5.8\times10^{-3}$ cm$^{-3}$ s$^{-1}$)

We checked the relationship between rainfall intensity and particle number concentration and they were not significantly related.

Considering the referee's comments, in the revised version, we have modified the discussions on the dependence of BWSCs on particle number concentrations prior to rainfall events in Line 366-379 and Figure S16.

Comment 10:
L341-342: How? Below-cloud wet scavenging is a first-order process. It should not depend on the number. However, it does depend on particle size. See the two comments above. The interpretation here seems incorrect.
**Response:** Thanks for the comment. Based on the two comments and responses above, we have modified the relevant discussions in section 3.4. Please see Line 357-379.

Comment 11:
L342-344: Unless the particle concentrations are extremely high, below-cloud scavenging should be removing particle number much faster than coagulation.
**Response:** Thanks for the comment. Based on the comments 8-9 and responses, we have modified the relevant discussions in section 3.4. Please see Line 357-379.

**References:**

Andronache, C., Grönholm, T., Laakso, L., Phillips, V., and Venäläinen, A.: Scavenging of ultrafine particles by rainfall at a boreal site: observations and model estimations, Atmos. Chem. Phys., 6, 4739-4754, 10.5194/acp-6-4739-2006, 2006.

Atlas, D., and Ulbrich, C. W.: Path- and Area-Integrated Rainfall Measurement by Microwave Attenuation in the 1–3 cm Band, Journal of Applied Meteorology and Climatology, 16, 1322-1331, 1977.

Blanco-Alegre, C., Castro, A., Calvo, A. I., Oduber, F., Alonso-Blanco, E., Fernández-González, D., Valencia-Barrera, R. M., Vega-Maray, A. M., and Fraile, R.: Below-cloud scavenging of fine and coarse aerosol particles by rain: The role of raindrop size, Quarterly Journal of the Royal Meteorological Society, 144, 2715-2726, 10.1002/qj.3399, 2018.

Cugerone, K., De Michele, C., Ghezzi, A., and Gianelle, V.: Aerosol removal due to precipitation and wind forcings in Milan urban area, Journal of Hydrology, 556, 1256-1262, 10.1016/j.jhydrol.2017.06.033, 2018.

Geng, T., Tong, H., Zhao, X., and Zhu, Y.: Effect of Wet Deposition on the Removal Efficiency of Particulate Matter in the Yangtze-Huaihe Region, Research of Environmental Sciences, 32, 273-283, 2019.

Greenfield, S.: Rain scavenging of radioactive particulate matter from the atmosphere, Journal of Meteorology, 14, 115-125, 1957.

Laakso, L., Grönholm, T., Rannik, Ü., Kosmale, M., Fiedler, V., Vehkamäki, H., and Kulmala, M.: Ultrafine particle scavenging coefficients calculated from 6 years field measurements, Atmospheric Environment, 37, 3605-3613, 10.1016/S1352-2310(03)00326-1, 2003.

Luan, T., Guo, X., Zhang, T., and Guo, L.: The Scavenging Process and Physical Removing Mechanism of Pollutant Aerosols by Different Precipitation Intensities, Journal of Applied Meteorolgical Science, 30, 279-291, 2019.

Pryor, S. C., Joerger, V. M., and Sullivan, R. C.: Empirical estimates of size-resolved precipitation scavenging coefficients for ultrafine particles, Atmospheric Environment, 143, 133-138, 10.1016/j.atmosenv.2016.08.036, 2016.

Roy, A., Chatterjee, A., Ghosh, A., Das, S. K., Ghosh, S. K., and Raha, S.: Below-cloud scavenging of size-segregated aerosols and its effect on rainwater acidity and nutrient deposition: A long-term (2009–2018) and real-time observation over eastern Himalaya, Science of The Total Environment, 674, 223-233, 10.1016/j.scitotenv.2019.04.165, 2019.

Seinfeld, J. H., and Pandis, S. N.: Atmospheric chemistry and physics: from air pollution to climate change, John Wiley & Sons, 2016.

Slinn, W. G. N.: Precipitation scavenging, Atmospheric Sciences and Power Production – 1979, chap. 11, 1983.

Wang, X., Zhang, L., and Moran, M. D.: Uncertainty assessment of current size-resolved parameterizations for below-cloud particle scavenging by rain, Atmos. Chem. Phys., 10, 5685-5705, 10.5194/acp-10-5685-2010, 2010.

Wang, Y., Zhu, B., Kang, H., Gao, J., Jiang, Q., and Liu, X.: Theoretical and observational study on below-cloud rain scavenging of aerosol particles, Journal of University of Chinese Academy of Sciences, 31, 306-313, 2014.

---

## Author Comment (AC2)

**Response to Reviewer #2**

The paper investigates possible sources of discrepancy in the below-cloud wet scavenging coefficient (BWSC), a metric related to the ability of rainfall to remove suspended particles, between observations and theoretical results. Past studies have highlighted that the BWSC inferred from measured particle number size distributions (PNSD) are larger than the theoretical ones, but sources of these discrepancies remain unclear. This paper uses long-term (i.e., 7 years, ~1800 days after data quality control) PNSD data collected at the SORPES station in eastern China to quantify BWSC in a polluted atmosphere and to investigate sources of discrepancies between observed and theoretical BWSC. Auxiliary to the analysis is the ERA5 and TRMM data that provide meteorological and rainfall information over the area, respectively. The authors select rainfall days above a certain intensity and duration and exclude extreme events and frontal passages that will significantly impact PNSD. This leads to 170 events being analyzed. Two categories of events are identified based on BWSC (high and normal events). Events are also classified based on changes in synoptic systems that may lead to a different role played by the BWSC. Pollutant concentrations (e.g., CO) are also used as tracers to track changes in air masses. The manuscript is well written, the results are innovative, and topic is of high relevance for Atmospheric Chemistry and Physics. However, the following concerns should be addressed to be considered suitable for publication.

**Response:** Thanks for providing helpful and constructive comments, which help us further improve the manuscript. The changes in the revised manuscript were highlighted in red color. Below is our point-by-point response to each comment.

Comment 1:

Few results are shown for the normal scavenging vs high scavenging events in terms of PNSD. Two specific events are shown in Fig 3, but it would be interesting to know more about the overall statistics. How do the two distributions differ on average? What is the variability/uncertainty in the PNSD and rainfall distribution over time for these two types of events over the analyzed period?

**Response:** Thanks for the comment and suggestion. As shown in Fig. R1, the average PNSD before rainfall for normal scavenging events and high BWSC events are similar. The particle number concentrations of normal scavenging events show a certain degree of reduction in all particle size bins, and the dominant particle diameter changes less. The particle number concentrations of the high BWSC events show more obvious reduction in most of particle size bins. The dominant particle diameter changes significantly from 100 nm to 50 nm, indicating the different source of particles due to the air masses changing. The variability in PNSD of the normal scavenging events is larger than that of the high BWSC events.

Considering the referee's comments, we have added more descriptions of the particle number size distribution for normal and high BWSC events in the Line 203-206 and Figure S1.

[Figure]

**Figure R1.** The PNSD before and after rainfall events for (a) normal scavenging events and (b) high BWSC events. The lines represent the median, 25th and 75th percentiles and the circles represent the average.

Comment 2:
More details should be provided in the methods on how the different events are classified in terms of BWSC and also in terms of the synoptic driving systems. Also for example, in Figure 6 it's not clear what the trajectory height means and how it is computed. Rainfall events may be synoptic driven or more locally generated. How are these differences accounted for?

**Response:** Thanks for the comment. The normal and high BWSC events separated in Fig. 2 were classified by analyzing the time series of PNSD for each rainfall event, as shown in Fig. 3. If there is a rapid and sudden decrease in particle number concentration in all size ranges near the end of the rainfall, it is classified as the high BWSC event. Otherwise, it is the normal event. The classification of the synoptic situation in section 3.2 is conducted using the subjective procedure. The synoptic situation type of each rainfall event is classified according to the geopotential height and wind fields on 850 hPa (Bei et al., 2016). When a trough moves from west to east, the synoptic situation is categorized as "westerly trough type" (16.6%); when a stationary front moves slowly at the SORPES station, the synoptic situation is categorized as "stationary front type" (31.3%); when a vortex weakens into the small trough at SORPES station, the synoptic situation is categorized as "vortex weakening type" (22.9%); when a vortex stays at SORPES station, the synoptic situation is categorized as "vortex type" (29.2%). The trajectory height refers to the maximum altitude of the 36-hour backward trajectory. The 36-hour backward trajectories are calculated by the HYSPLIT (Hybrid Single-Particle Lagrangian Integrate Trajectory) model driven with the GDAS (Global Data Assimilation System) reanalysis data.

Nanjing is strongly affected by the East Asian monsoon and is located in the lower reaches of the Yangtze River, which can reflect the main rainfall characteristics of this area (Wan et al., 2012). The accumulated precipitation and rainfall frequency are high in Nanjing, especially from June to August (Fig. 1a). The rainfall in this area is often

caused by synoptic situations such as fronts, vortexes and tropical cyclones (Luo et al., 2016; Yongguang et al., 2008). Short-term heavy rainfall caused by local convection also occurs from time to time, accompanied by very high rainfall intensity (≥20 mm/h) and drastic changes in meteorological conditions (Shen et al., 2015; Yang et al., 2015).

In section 2.2, we have screened for rainfall events, including rainfall duration and changes in meteorological conditions. The screening procedure removed the events where drastic meteorological condition changes occurred. Therefore, the 170 rainfall events in this study are mainly synoptic driven.

Considering the referee's comments, we have added more descriptions and discussions in the revised manuscript. Please see Line 223-228, Line 295-296, Line 311-312 and Line 139-141.

Comment 3:
The dependence of BWSC on the pre-existing particle number concentrations is very interesting, but more details are needed. Does this relationship vary as a function of the particle size as well or is it constant from the smallest to the largest diameters? Does the pre-existing PNSD play a role in dictating BWSC or is the total particle number the key driver?

**Response:** Thanks for the comment.

[Figure]

**Figure R2.** Below-cloud wet scavenging coefficients (median over particle sizes 10-500 nm) for high BWSC events and normal events as the function of (a) rainfall

intensity and (b) particle number concentration prior to rainfall events. The dots represent the median and the error bars represent the upper and lower quartiles. Below-cloud wet scavenging coefficients in each size bin for normal events as the function of (c) rainfall intensity and (d) particle number concentration prior to rainfall events.

In the revised manuscript, we presented the BWSCs for each size bin (Fig. R2c, d). As shown in Fig. R2c, the dependence of BWSC on rainfall intensity for particles below 100 nm (i.e. ultrafine particles) is similar with that of total particles, since the ultrafine particles dominate the particle number concentration at SORPES. The BWSCs are low for the particles larger than 100 nm due to the exist of "Greenfield" gap (Greenfield, 1957). Nevertheless, the increase in BWSC with increasing rainfall intensity can also be found for the particles larger than 100 nm. The dependence of BWSC on particle number concentration prior to rainfall event is obvious for ultrafine particles as well (Fig. R2d). In general, although the BWSCs are related to the particle size, the dependences of BWSCs on rainfall intensity as well as particle number concentration is uniform over all the size bins.

[Figure]

**Figure R3.** The variation of particle number concentration with time during rainfall under theoretical conditions (a) without and (b) with the stable source. ($\lambda=1\times10^{-5}$, $A=5.8\times10^{-3}$)

Theoretically, below-cloud wet scavenging is a 1st-order loss process and should not have the relationship with the particle number concentration prior to rainfall event. The basic equation of variation in the particle number concentration $c(d_p)$ due to rainfall scavenging is described by Eq. (1) (Seinfeld and Pandis, 2016):

$$\frac{dc}{dt} = -\lambda c \tag{1}$$

$\lambda$ is the scavenging coefficient (i.e. BWSC). As shown in Fig. R3a, the particle number concentration varies exponentially with time if no other processes (other sources or sinks of particles) are present during the rainfall event. The slope of the line is the scavenging coefficient ($\lambda$) and does not vary with the particle number concentration

prior to rainfall ($c_0$).

However, in the real ambient environments, other processes such as particle formation, primary emissions, coagulation, etc. cannot be excluded. For example, if there is a stable source of particles, the variation of particle number concentration can be described by:

$$\frac{dc}{dt} = -\lambda c + A. \tag{2}$$

A is the formation rate of the particles due to the stable source. The variation of particle number concentration based on Eq. (2) is shown in Fig. R3b by assuming consistent BWSC and formation rate.

In field measurements, it is common to use Eq. (3) to calculate the BWSC:

$$\lambda = -\frac{1}{t_1 - t_0} \ln\left(\frac{c_1}{c_0}\right). \tag{3}$$

$c_0$ and $c_1$ are the median particle number concentrations before ($t_0$) and after ($t_1$) the rainfall event, respectively. Assuming same duration of rainfall event, the BWSCs can be calculated based on Eq. (3) for different initial particle number concentration (dash lines in Fig. R5b). Although the actual BWSC is consistent, the calculated BWSC can increase with the increasing initial particle number concentration. Therefore, the relationship between BWSC and particle number concentration prior to the rainfall event can be caused by other processes except for below cloud wet scavenging. In real ambient environments, other processes perturb the calculation of BWSC when using the BWSC calculation method as described by Eq. (3). We thereby highlight the need for further research on BWSC calculation methods based on field measurements.

Considering the referee's comments, in the revised version, we have modified the discussions on the dependence of BWSCs on particle number concentrations prior to rainfall events in Line 357-379 and Figure 8 and Figure S16.

Comment 4:
How generalizable these results are? The authors could contextualize the emission regimes, the typical PNSD, the type of air mass changes, with those observed in other studies and comment on which regions the BWSC could be impacted by some of the identified drivers.
**Response:** Thanks for the comment and suggestion.

In urban environment, most of the observed studies show that the observed BWSCs are much higher than the theory. Zhao et al. (2015) studied below-cloud wet scavenging of aerosol in Lanzhou. Lanzhou is one of the megacities in western China and has been suffering from severe air pollution due to rapid urbanization and industrialization (Zhang et al., 2017). The distribution pattern of BWSC between 40-400 nm is very similar to the high BWSC event without the significant "greenfield gap" (Greenfield, 1957). Xu et al. (2019) conducted field measurements in Beijing and used multiple methods to calculate BWSC. The rainfall event studied by Xu et al. (2019) clearly showed a rapid and sudden decrease in particle number concentration in all size ranges

near the end of the rainfall, which is similar with the phenomenon illustrated in Fig. 3b. By investigating the weather conditions and backward trajectories of the below-cloud scavenging events observed in those studies, we suggest that the air mass changing is one of the non-negligible reasons of high BWSC event observed in the studies by Zhao et al. (2015) and Xu et al. (2019).

Relatively high BWSCs were also observed in clean environment, such as boreal forest in Finland (Laakso et al., 2003). We investigated the rainfall events observed at SMEAR II station in boreal forest of Finland and found the similar high BWSC events, which belong to type iv high BWSC event. As shown in Fig. R4, the SMEAR II station was located on the southern side of the vortex, and the vortex showed no clear movement. The backward trajectories of air masses were from the Baltic Sea. The rapid decrease in particle number concentration in all size ranges was observed at the end of the rainfall, which is caused by the influence of marine air masses.

In summary, the high BWSC events (a rapid decrease in particle number concentration in all size ranges near the end of the rainfall) mentioned in this study can be found not only in urban environment but also in clean natural environments. We thereby highlight all the below cloud scavenging studies based on the field measurements need to consider the influence of air mass changing. Considering the referee's comments, we have added more discussions about the generalizability of the conclusion in this study in the revised manuscript. Please see Line 395-396.

[Figure]

**Figure R4.** Distributions of winds and geopotential heights at 850 hPa (a) before, (b) at, and (c) after the moment of sudden decrease in particle number concentration. (d) Top to bottom panels are time series of wind, ambient temperature, RH, precipitation, and particle number size distribution. (e) The backward trajectory before, at, and after the moment of sudden decrease in particle number concentration.

**References:**

Bei, N., Li, G., Huang, R. J., Cao, J., Meng, N., Feng, T., Liu, S., Zhang, T., Zhang, Q., and Molina, L. T.: Typical synoptic situations and their impacts on the wintertime air pollution in the Guanzhong basin, China, Atmos. Chem. Phys., 16, 7373-7387, 10.5194/acp-16-7373-2016, 2016.

Greenfield, S.: Rain scavenging of radioactive particulate matter from the atmosphere, Journal of Meteorology, 14, 115-125, 1957.

Laakso, L., Grönholm, T., Rannik, Ü., Kosmale, M., Fiedler, V., Vehkamäki, H., and Kulmala, M.: Ultrafine particle scavenging coefficients calculated from 6 years field measurements, Atmospheric Environment, 37, 3605-3613, 10.1016/S1352-2310(03)00326-1, 2003.

Luo, Y., Wu, M., Ren, F., Li, J., and Wong, W.-K.: Synoptic Situations of Extreme Hourly Precipitation over China, Journal of Climate, 29, 8703-8719, https://doi.org/10.1175/JCLI-D-16-0057.1, 2016.

Seinfeld, J. H. and Pandis, S. N.: Atmospheric chemistry and physics: from air pollution to climate change, John Wiley & Sons2016.

Shen, C., Yan, T.-b., Liu, D.-q., and Li, J.: Characteristics of short-time heavy precipitation from 2008 to 2012 in Nanjing, Journal of Meteorology and Environment, 31, 28-33, 2015.

Wan, F., Yuan, H., Song, J., and Wang, Y.: Research on precipitation forecasts in Nanjing City, Journal of Nanjing University, 48, 513-525, 2012.

Xu, D., Ge, B., Chen, X., Sun, Y., Cheng, N., Li, M., Pan, X., Ma, Z., Pan, Y., and Wang, Z.: Multi-method determination of the below-cloud wet scavenging coefficients of aerosols in Beijing, China, Atmos. Chem. Phys., 19, 15569-15581, 10.5194/acp-19-15569-2019, 2019.

Yang, S., Anyun, S., Haiying, W., and Dongping, Y.: Structure analysis on the meso-$\beta$ scale convective system in a local torrential rain of Nanjing, Journal of the Meteorological Sciences, 35, 791-798, 10.3969/2014jms.0068, 2015.

Yongguang, Z., Jiong, C., Guoqing, G. E., Yanfang, H., and Chunxi, Z.: Review on the Synoptic Scale Meiyu Front System and Its Synoptics Definition, Acta Scientiarum Naturalium Universitatis Pekinensis, 44, 157-164, 2008.

Zhang, X., Zhang, Y., Sun, J., Zheng, X., Li, G., and Deng, Z.: Characterization of particle number size distribution and new particle formation in an urban environment in Lanzhou, China, Journal of Aerosol Science, 103, 53-66, https://doi.org/10.1016/j.jaerosci.2016.10.010, 2017.

Zhao, S., Yu, Y., He, J., Yin, D., and Wang, B.: Below-cloud scavenging of aerosol particles by precipitation in a typical valley city, northwestern China, Atmospheric Environment, 102, 70-78, 10.1016/j.atmosenv.2014.11.051, 2015.